# A Control Alternative for the Hidden Enemy in the Wine Cellar

**Rubén Peña [1], Renato Chávez [2], Arturo Rodríguez [3]  and María Angélica Ganga [1,*]**

[1]   Departamento en Ciencia y Tecnología de los Alimentos, Facultad Tecnológica, Universidad de Santiago de Chile, Santiago 9170201, Chile; ruben.pena@usach.cl

[2]   Departamento de Biología, Facultad de Química y Biología, Universidad de Santiago de Chile, Santiago 9170201, Chile; renato.chavez@usach.cl

[3]   Departamento de Tecnologías Industriales, Facultad Tecnológica, Universidad de Santiago de Chile, Alameda 3363, Estación Central, Santiago 9170201, Chile; arturo.rodriguez@usach.cl

*   Correspondence: angelica.ganga@usach.cl; Tel.: +56-2-2718-4509

**Abstract:** *Brettanomyces bruxellensis* has been described as the principal spoilage yeast in the winemaking industry. To avoid its growth, wine is supplemented with $SO_2$, which has been questioned due to its potential harm to health. For this reason, studies are being focused on searching for, ideally, natural new antifungals. On the other hand, it is known that in wine production there are a variety of microorganisms, such as yeasts and bacteria, that are possible biological controls. Thus, it has been described that some microorganisms produce antimicrobial peptides, which might control yeast and bacteria populations. Our laboratory has described the *Candida intermedia* LAMAP1790 strain as a natural producer of antimicrobial compounds against food spoilage microorganisms, as is *B. bruxellensis,* without affecting the growth of *S. cerevisiae.* We have demonstrated the proteinaceous nature of the antimicrobial compound and its low molecular mass (under 10 kDa). This is the first step to the possible use of *C. intermedia* as a selective bio-controller of the contaminant yeast in the winemaking industry.

**Keywords:** antimicrobial peptides; biocontrol; *Brettanomyces bruxellensis*; *Candida intermedia*; wine; off-flavors

## 1. Introduction

Phenol derivatives have been identified as one of the volatile components which provide a pleasant aroma to wine when produced in low concentration [1,2]. The most important molecules that belong to this group are 4-vinylphenol, 4-vinylguaiacol, 4-ethylphenol, and 4-ethylguaiacol [1,3]. Nevertheless, there are threshold values for these components; thus, an increase of the concentration produces an off-flavor in wine. Some authors have established that concentrations over 620 µg/L of 4-ethylphenol produced aromas related to "phenol", "barn", "horse sweat", "leather", "varnish", among others [2,3], which causes important economic losses for the industry [4,5]. However, concentrations under 400 µg/L, 4-ethylphenol contribute to the aromatic complexity of the product, providing notes of "spices", "leather", and "smoke" which are valued by most wine consumers [2].

## 2. Production of Phenolic Derivatives

The precursors of the phenol derivatives are the phenolic acids or hydroxycinnamic acids (*p*-coumaric, ferulic, caffeic, and sinapinic acids). These compounds are naturally found in grapes and in vegetal tissues conjugated with tartaric acid as a natural part of the grape peel [2,6]. The hydroxycinnamic acids can be released during winemaking [2,6]. Besides, some microorganisms

that are in the must release enzymes which would also help to release these acids [2]. It has been described that hydroxycinnamic acids would exert an inhibitory effect on the growth of microorganisms, due to imbalances produced in the cell medium. In this context, ferulic and *p*-coumaric acids would exert the most inhibitory effects in yeasts [7]. For this, microorganisms that are able to ferment vegetable products show enzymatic activity, which would allow these compounds to metabolize into less toxic ones [1,8]. The most studied and characterized pathway to transform these hydroxycinnamic acids into volatile phenols corresponds to the sequential action of two enzymes; first, the action of phenolic acid decarboxylase transforms hydroxycinnamic acids into vinyl derivatives and, posteriorly, these compounds are reduced to ethyl derivatives by the action of a vinyl reductase [1,8,9].

The presence of phenolic acid decarboxylase activity has been related to *Bacillus* and *Lactobacillus* bacteria and, *Saccharomyces* and no-*Saccharomyces* yeasts [1,3]. On the other hand, it has been described that only yeasts from the genus *Brettanomyces/Dekkera*, *Kluyveromyces*, *Candida*, and *Pichia* would be able to generate ethyl derivatives, even though only *Pichia* and *Brettanomyces/Dekkera* species could produce important quantities that surpass the sensorial threshold [10,11]. The presence of acids in the must, which are mainly esterified with tartaric acid, has been described. Thus, the generation of their free forms is dependent on the presence of microorganisms, which have enzymes with cinnamoyl esterase activities [2]. Among hydroxycinnamic acids present in the must, there are the *p*-coumaric, ferulic, and caffeic acids; being *p*-coumaric found in greater quantity [2,3].

## 3. *Brettanomyces/Dekkera* as a Wine Spoilage Yeast

In this genus has been described the anamorphs *B. bruxellensis*, *B. anomalus*, *B. custersianus*, *B. naardenensis*, and *B. nanus*, with teleomorphs existing for the first two species, *Dekkera bruxellensis* and *Dekkera anomala* [12]. *B. bruxellensis* has been described as being mainly responsible for off-flavor production in wine worldwide. In the case of *B. bruxellensis*, it presents slow growth, fermentative and oxidative metabolism, consumption of several sugars, production of acetic acid under aerobic conditions, and natural resistance to the antifungal compound cyclohexamide [2,6,13]. From the enological point of view, *B. bruxellensis* is recognized for its high tolerance to ethanol and its capacity of surviving in environments which lack nutrients and have a low pH, allowing persistent proliferation in winemaking processes [6,12]. Nevertheless, the distinct characteristic of this species is the capacity of transforming hydroxycinnamic acids present in the must into phenolic derivatives, which affect the organoleptic quality of wine [1,3,6,8–11].

Contamination by *B. bruxellensis* can occur during the winemaking process. Nevertheless, its proliferation is favored during the ageing period in barrels [2]. In this context, *B. bruxellensis* can settle in the pore microstructure of the wood [13]. Besides, it has been described that this yeast can decompose cellobiose, so it can supply its nutritional necessities from wood and keep metabolically active in this structure during several generations [2,13]. These yeast features increase the risk to transform the barrels into carriers and transmitters of contamination. Therefore, its proliferation is very difficult to eradicate.

## 4. Control of *B. bruxellensis*

For many years the wine industry has looked for tools to eradicate contaminant microorganisms in the fermentation and ageing processes of wine in barrels. This problem has been studied chemically, through anhydrous sulfide addition ($SO_2$), as a potassium metabisulfite form [14]. This compound is frequently used as a preservative, due to its antimicrobial, antioxidant and stabilizing properties to the final product [4]. $SO_2$ supplement is carried out over must to decrease its natural microbiological charge. However, it is common to repeat its addition after alcoholic fermentation and during the ageing in barrels, with the aim of avoiding the growth of spoilage microorganisms during this process [4,14]. The anhydrous sulfide is found normally in chemical equilibrium between its molecular form ($SO_2 \cdot H_2O$) and the bisulfite anion ($HSO^{3-} + H^+$). The molecular $SO_2$ can diffuse into the cell cytoplasm and dissociate now between bisulfite ($HSO^{3-} + H^+$) and sulfite ($SO_3^{2-} + H^+$) [14]. This dissociation

produces a sustaining increase of the concentration protons in the cytoplasm, which generates a rapid acidification and an abrupt redox imbalance. Additionally, it has been determined that the sulfite anion ($SO_3^{2-}$) is highly reactive and produces the inactivation of several metabolites and cell enzymes. Besides the penetration of molecular $SO_2$ to yeasts, cytoplasm produces the immediate inhibition of the glyceraldehyde-3-phosphate dehydrogenase enzyme, interrupting glycolysis and NADH regeneration, allowing ATP depletion [14]. In the case of *S. cerevisiae,* the presence of *SSU1* gene has been described, which codifies for a $SO_2$ efflux pump, making this yeast resistant to this compound and able to survive to generate the alcoholic fermentation [15,16]. Among the physiological and molecular studies carried out on *B. bruxellensis,* it has been determined that this yeast shows strain-dependent resistance to $SO_2$ [4,17]. This phenomenon has been related to the presence of an ortholog gene to *SSU1* in the genome of *B. bruxellensis* AWRI1499, which may affect this strain tolerance to $SO_2$ [16]. Another study has shown the tolerance profiles to this compound in 108 *B. bruxellensis* strains, obtained from different geographical origins. The results showed that 19 strains do not tolerate 0.1 mg/L $SO_2$, 29 grow with 0.1 mg/L, 42 tolerate 0.2 mg/L, 16 tolerate 0.4 mg/L, and two tolerate over 0.6 mg/L $SO_2$ [4]. This phenomenon is relevant in the industry because it has been reported that $SO_2$ can be a potentially harmful agent for human consumption, due to it producing irritation of the gastric mucosa, dizziness, headache and, in susceptible individuals, it can cause allergic and severe asthmatic crisis [18].

The use of food industry sanitizers, as alkaline detergents and iodophors, has low use in the wine industry due to the complex geometry of their machines (bottling machines, valves, etc.) or to the low access for cleaning of the superficies [13]. On the other hand, dimethyl dicabonate (DMDC) is a preservative authorized to be used in winemaking in some countries; its efficiency depends on the strain, temperature, ethanol concentration, and pH. DMDC is rapidly hydrolyzed, the effect done instantaneously in must or wine; however, its use in large volumes has low effectiveness. For this reason, DMDC is recommended to be used in the presence of molecular sulfur dioxide [19]. Another compound studied to reduce the *B. bruxellensis* population in wine has been chitosan, a natural polymer obtained from the exoskeletons of crustaceans. At laboratory level, the studies show a control on the growth of *Brettanomyces*; however, at industrial level there is not complete eradication, with there being the fungistatic effect limited in time [20].

So, the wine industry is looking for new technological solutions, which allow eradication of this yeast in the fermentation and ageing processes of wine in barrels. Biotechnological investigation has provided several physics strategies to avoid contamination by *B. bruxellensis*. Thus, some works study the exposition of contaminated must and wine to pulses of a defined electric field. It was reported that the application of a pulse of 29 kV/cm (186 kJ/kg) reduces the viability of contaminant bacteria and yeasts by 99.9%, such as *Lactobacillus hilgardii*, *Lactobacillus plantarum*, *D. anomala,* and *B. bruxellensis* [21]. On the other hand, the study of the treatment of contaminated barrels with deionized water at different temperatures determines that submerging barrels during 19 min in water at 60 °C reduces the growth of four *B. bruxellensis* strains in eight logarithmic cycles [22]. Furthermore, the application of hydrostatic pressure on the growth of strains *B. bruxellensis* in synthetic must, at different pH and ethanol concentrations, was studied. The results showed that one minute treatment at a pressure of 300 MPa totally reduces the viability of contaminant yeast [23]. Nevertheless, these physics strategies have not been effectively incorporated by the industry due to their low technical and economic feasibility for implementation. From a biological point of view, several authors have focused on the identification and characterization of natural killer toxins with antifungal properties. In this context, the first report of a killer toxin that had an effect on the growth of *B. bruxellensis* was found in the non-*Saccharomyces* yeasts, *Pichia anomala* (Pikt) and *Kluyveromyces wickerhamii* (Kwkt). In this study, it was determined that both toxins showed the capacity of modulating the proliferation of a contaminant yeast in wine for 10 days [24]. However, after that period, the bio-controlled efficiency of toxins on the yeast proliferation is not described. Posteriorly, the use of PMKT2 toxin of *Pichia membranefaciens* on *S. cerevisiae* and *B. bruxellensis* in must was described. Here it was determined that PMKT2 is an effective bio-controller of the contaminant yeast, but it also affects the growth of the

fermentative yeast [25]. Therefore, it is not considered a good tool for the industry. Later, the production of a killer toxin secreted *Ustilago maydis* has been described [26]. This work demonstrated, using vinification assays, that the toxin affects the growth of *B. bruxellensis*, while *S. cerevisiae* shows complete resistance. Besides, it was observed that supplementing toxins in the fermentation and ageing conditions of wine in barrels reduces the content of 4-ethylphenol produced by the contaminant yeast significantly. This result demonstrates its effective reduction of volatile phenols that cause the aromatic default. Nevertheless, 17 *B. bruxellensis* strains used in the study show relative sensitivities, with 10 of them being low sensitivity [26]. This phenomenon constrains the use of the toxin killer *U. maydis* as a general bio-control strategy against *B. bruxellensis*, since not all strains would be susceptible, so that the accumulation of volatile phenols in wine would be at random.

Additionally, the production of toxins CpKT1 and CpKT2 for *Candida pyralidae* YWBT Y1140 strain has been described. In this study, it was demonstrated that toxins present a molecular mass over 50 kDa and stability in an acidic pH, high alcoholic degree, and different sugar concentrations. Nevertheless, its effectiveness was studied in a laboratory medium. In white and red wine, the toxins showed that only seven strains, out of 15 *B. bruxellensis* strains studied, present sensitivity in the wine matrix [27]. Later, the same authors demonstrated that toxins CpKT1 and CpKT2 can cause damage to cell walls in *B. bruxellensis* sensitive strains [28].

Several investigation groups have focused on the search of antimicrobial peptides (AMPs), which have been studied as different microorganism agent controllers at clinical or industrial importance [27].

## 5. Antimicrobial Peptides and their Antifungal Action Mechanisms

Antimicrobial peptides (AMPs) are molecules which are found in a broad range of organisms (from prokaryotic cells to human beings) and constitute the first line of defense against potentially pathogenic organisms in multicellular organisms. However, some microorganisms are able to produce AMPs with the purpose of ensuring survival [29]. Generally, AMPs show relative length (below 100 amino acid residues) and can differ in sequence. Nevertheless, they share, as a distinctive characteristic, the presence of amino acid residues charged to physiological pH and non-polar residues, which determine its amphipathic nature [30–32]. The analysis of the tridimensional structure of different AMPs has shown that these can be linear or adopt α-helical or β-laminar conformation, in which charged and hydrophobic residues are aligned in the opposed faces, allowing its water solubility [30,31]. Due to the high number of identified peptides "The Antimicrobial Peptide Database (APD3)" was generated (http://aps.unmc.edu/AP/main.php). This database is a library in which peptides are classified according to origin, sequence, activity, and structural or physiochemical properties [33]. To date, APD3 contains peptides obtained from different kingdoms, from which 13 are produced by species corresponding to *Fungi.* From them, six peptides were identified in *Fungi* such as *Aspergillus giganteus*, *Aspergillus clavatus*, *Aspergillus niger*, *Penicillium chrysogenum,* and *Pseudoplectania nigrella* [34–39]. Other authors have described the genus *Trichoderm* can produced antimicrobial peptides of no-ribosomal synthesis named peptaibols. They are characterized by a length between seven and 20 amino acid residues, from which a high proportion corresponds to no-proteinogenic amino acids, such as isovaline and α–aminoisobutyric. Furthermore, they show acetylation at the N-terminal and amino alcohol at the C-terminal [40]. To date, 317 peptaibols have been described, which have been stored according to origin, sequence, and crystallographic structure at the Peptaibol database (http://peptaibol.cryst.bbk.ac.uk/home.shtml) [41].

One of the main aspects in the study of AMPs and peptaibols has been the determination of their antifungal action mechanism. When the antifungal effect of a peptide produced by *Aspergillus giganteus* was studied, it was determined that their action mechanism is related to cell wall permeabilization [36]. Further, through immunofluorescence experiments, it was determined that this AFP (antifungal protein) is exclusively located in the plasmatic membrane, so that its inhibitory effect would be related to the joining and destabilization of this structure [36]. This mechanism would be similar to what was described for identified peptaibols in different species of genus *Trichoderma* [41]. The peptaibol

alamethicin (isolated from culture medium of *Trichoderma viride*) showed amphiphilic characteristics and it is strongly absorbed by natural and synthetic membranes and, consequently, generated cell lysis [42]. On the other hand, different computing simulations were carried out by using the sequence, tridimensional structure, physicochemical properties of several AMPs, and alamethicin peptaibol to potentially determine its action mechanism. It was determined that these attach to the outer face of the plasmatic membrane and its accumulation produces disorganization of the phospholipid bilayer, favoring emerging pores. These pores would allow ion efflux (mainly potassium) from intracellular, which causes an ionic gradient imbalance, oxygen reactive species production (ROS), and the subsequent cell death [43]. In this context, cell membrane permeabilization is not the only action mechanism related to AMPs. When an antifungal peptide secreted by *P. chrisogenum* (PgAF) was characterized, it was determined that it corresponds to a peptide whose length is 55 amino acid residues, rich in cysteine, and 25% hydrophobic amino acid residues [35]. Likewise, by sequence alignment, it has been determined that PgAF presents a 42% sequence identity to an antifungal peptide from *A. giganteus* [36], 37% to a novel antifungal peptide from *A. niger* (AnAFP) [34], and 100% to *Penicillium nalgiovense* antifungal protein (NAF) [35]. Using proteins with antifungal activity from *Penicillium chrysogenum,* it was determined that these proteins produce cell wall disorganization and membrane permeabilization. This produces a great loss of turgidity and ionic gradient due to the rapid potassium efflux. The authors also reported an effect on the growth tips of hyphae and generation of reactive oxygen species (ROS) [35]. This observation was the first evidence of antimicrobial peptides and intracellular toxicity by ROS generation. Another piece of research has studied the antifungal effect of a hexapeptide derived from peptide PAF of *P. chrysogenum*, named PAF26. This lineal hexapeptide (sequence RKKWFW) presents two well defined functional motives. The first, located at the N-terminal (RKK), corresponds to a cationic domain (net charge +3), while the second, located at the C-terminal (WFW), corresponds to a hydrophobic domain [44]. By fluorescence microscopy, it was determined that PAF26 is internalized by *Aspergillus fumigatus*, *Neurospora crassa,* and *S. cerevisiae*, exerting its antifungal action in the intracellular space [31]. Later, these authors proposed an action mechanism for PAF26 using, as models, *N. crassa* and *S. cerevisiae*. They observed two particular sub-mechanisms, which are directly related to the concentration of the hexapeptide in the medium. When concentrations between 2.5–5.0 μM PAF26 are applied, it was determined that the peptide interacts with natural negative charges of the cell wall. Once in contact with the membrane, PAF26 is internalized by the cell via generation of endosomes, producing the accumulation and expansion of vacuoles. After, by active transport of vacuoles, PAF26 is released into the cytoplasm where it produces permeabilization of membranes, allowing the release of mitochondrial ROS. Thus, cell death would be related to the oxidative stress of DNA, oxidative damage to the membrane level, and homeostatic intracellular imbalance [31]. In the second mechanism, when PAF26 concentration exceeds 20 μM, the hexapeptide translocation occurs through the cell membrane, followed by ROS generation and the subsequent oxidative damage on membranes and DNA. This effect produces cell death similar to what was described for low concentrations of PAF26 [31]. These mechanisms demonstrate that cell wall disorganization, membrane permeabilization, and cell internalization of AMPs produced by *Fungi* produce a redox imbalance in the target cell, whose final consequence is cell death.

## 6. Antimicrobial Peptides as a Contaminant Bio-Control Tool in the Winemaking Industry

Regarding the study of antimicrobial peptides as biocontrol contaminant microorganisms in the winemaking industry, the effect of synthetic fragments built from antimicrobial peptides produced by *P. chrysogenum* [35] and from the antimicrobial bovine peptide named Lactoferricin [45] has been analyzed. All peptides affected the growth of *B. bruxellensis*, *Cryptococcus albidus*, *Pichia membranifaciens*, *Zygosaccharomyces bailii,* and *Zygosaccharomyces* in a laboratory medium. Nevertheless, these peptides also affected (in less proportion) the growth of the fermentative yeast *S. cerevisiae* [46]. Posteriorly, the antifungal effect of the synthetic peptide LfcinB$_{17-31}$ on the growth of *B. bruxellensis* in a laboratory

medium, must, and white wine was studied, determining that this affected the growth in all media, and its action mechanism is related to the interaction and penetration of LfcinB$_{17\text{-}31}$ into the cell cytoplasm [46].

Actually, *S. cerevisiae* CCMI885 has been the only yeast in which the production of AMPs has been described. The production of antifungal compounds in a protein fraction between 2–10 kDa affected the growth of native wine yeast isolates of *B. bruxellensis*, *Hanseniaspora uvarum*, *Hanseniaspora guilliermondii*, *Candida stellata*, *Kluyveromyces thermotolerans*, *Kluyveromyces marxianus,* and *Torulaspora delbrueckii* [47]. Later, by characterization of these peptides through an Electrospray Ionizacion Mass Spectometry (ESI-MS), it was determined that it produces two peptides of molecular mass close to 1.6 kDa, which show a high sequence identity with isoforms one and two/three of the enzyme GAPDH [48]. Nevertheless, these peptides do not produce the complete inhibition of *B. bruxellensis* in a laboratory medium. Therefore, the surviving yeasts may continue producing the aroma default in a wine matrix [49]. Posteriorly, the authors studied the antifungal action mechanism of the characterized peptides, demonstrating that they produce disruption in the cell wall integrity in *H. guilliermondii* [50]. Finally, when synthetic isoforms of this AMP were produced, it was observed that they are not as effective as natural peptides [51]. By assessing the biochemical characteristics of AMPs and their antifungal action mechanisms, it can be said that the use of antimicrobial peptides for biocontrol of spoilage yeasts in the winemaking industry could be an effective tool. Our work group described the antimicrobial activity from several strains of yeasts isolated from winemaking environments, among them was *C. intermedia* LAMAP1790, which has antibacterial activity against the food pathogens *Escherichia coli*, *Listeria monocytogenes,* and *Salmonella typhimurium* [52]. Posteriorly, it was demonstrated that the *C. intermedia* LAMAP1790 strain affects the growth of the *B. bruxellensis* LAMAP 2480 strain, determining that the antifungal is released to the culture medium (Figure 1). Moreover, it was demonstrated that this antifungal does not affect the growth of the fermentative yeast *S. cerevisiae*, it being the most important species from the enological point of view [53].

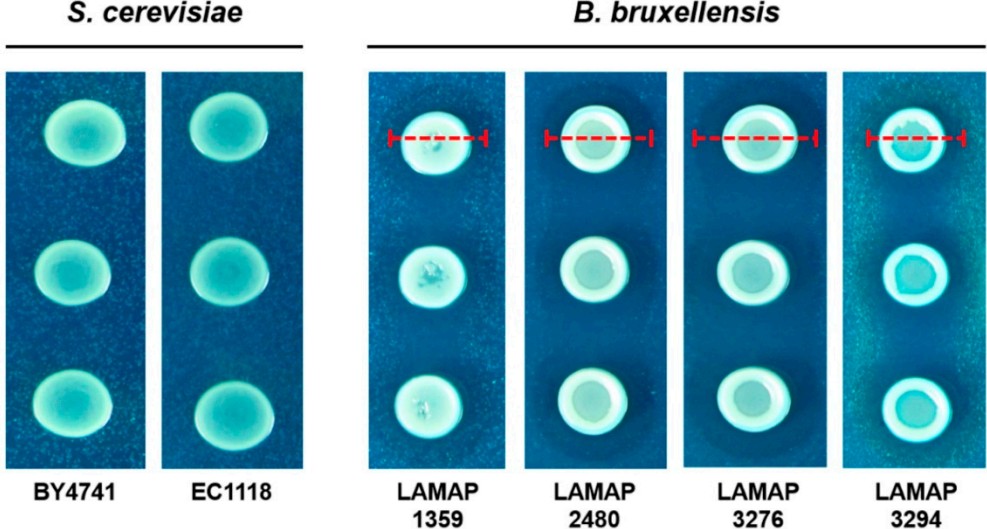

**Figure 1.** Semi-quantitative assessment of the antifungal action of *C. intermedia* LAMAP1790 on *S. cerevisiae* and *B. bruxellensis*. Each column corresponds to an inoculated strain in agar, while *C. intermedia* was inoculated as a drop in the layer three times. The antifungal capacity was quantified by measuring the diameter of the inhibition halo generated around *C. intermedia* LAMAP1790 (represented with red dotted lines). Left to right columns: *S. cerevisiae* BY4741, *S. cerevisiae* EC1118, *B. bruxellensis* LAMAP1359, *B. bruxellensis* LAMAP2480, *B. bruxellensis* LAMAP3276, and *B. bruxellensis* LAMAP3294 [51].

To determine whether the antifungal compound showed antifungal properties in a liquid medium, a viability assay was carried out by the exposition of the studied strains on culture sterile supernatant of *C. intermedia*. By comparing the counts of both strains of *S. cerevisiae*, it was observed that there are

not statistically significant differences in their growth and cell viability, the conclusion being it was not affected by the exposition to supernatant [53]. Nevertheless, by assessing the counts of *B. bruxellensis*, it was determined that there was not growth of *B. bruxellensis* LAMAP1359, *B. bruxellensis* LAMAP2480, and *B. bruxellensis* LAMAP3276 strains, and the *B. bruxellensis* LAMAP3294 strain showed statistically low viability. These results confirmed what was observed in the semi-quantitative assay (Figure 1), it being possible to demonstrate that the released compound to the culture medium, by *C. intermedia*, shows fungicidal activity against *B. bruxellensis*, without affecting *S. cerevisiae* [53].

To define the nature of the antifungal compound, an assay was carried out, in which a fraction of the supernatant was treated at 100 °C for 10 min and its antifungal capacity through viability of *B. bruxellensis* LAMAP2480 post-exposition was analyzed. Culture medium was used as a control, from which a fraction was submitted to thermic treatment [53]. These results show that when supernatant is kept at 4 °C, the antifungal activity was not affected, since the growth of *B. bruxellensis* was decreased by four logarithmic orders. Nevertheless, when supernatant is kept at 100 °C, its antifungal activity decreased significantly [53]. Later, supernatant was concentrated 100 times, and the total proteins were fractionated by ultrafiltration according to their molecular mass. Thus, two fractions were obtained, which contained proteins upper and lower than 10 kDa, respectively. It was demonstrated that the fraction lower than 10 kDa had antifungal activity and it disappeared in the presence of protease, showing the proteinaceous nature of the compound with antifungal activity. So, this is the first report of the production of peptides with antifungal capacity in a yeast different from *S. cerevisiae* [53].

## 7. Action Mechanisms of AMPs

Despite the knowledge obtained from the use of AMPs and their antimicrobial action mechanisms, to date it is not clear why some yeasts are resistant to AMPs and/or which mechanisms are used by the yeast producer to protect itself from its own antifungal action mechanism. For this reason, resistant mechanisms to AMPs in bacteria have been studied, determining that these can modify certain characteristics of their wall cell and membrane to protect themselves from the action of peptides [29]. Thus, it has been determined that *Staphylococcus aureus* transports alanine and lysine towards its wall to reduce negative net charge of teichoic acids; therefore, generating electrostatic repulsion with cationic AMPs. Additionally, it was determined that there is a positive correlation between the increase of membrane proteins with the resistant antimicrobial peptides in *Yersinia enterocolitica* [54].

To date, the only known resistant mechanisms to AMPs have been described in the pathogen yeast *C. albicans*. This yeast can defend itself from the attack of the antimicrobial peptide salivary histatin-5, which has been described as a relevant part of innate immunity in humans [55]. It has been described that *C. albicans* can defend against this peptide through three different mechanisms. In the first mechanism, this yeast is able to inactivate histatin-5 through secretion of proteases Sap9 and Sap10, which are able to digest antimicrobial peptide, besides degrading tissue of the host, which allows it to avoid the innate defense system and colonize the human oral cavity [56,57]. On the second mechanism, *C. albicans* secretes a glycoprotein named Msb2, which is able to join free AMPs in its glycoside realm, reducing its effective concentration and detoxifying the medium [57]. Finally, in the third mechanism, it has been described that *C. albicans* is able to expel the histatin-5 peptide from its cytoplasm, using as efflux pump Flu1, which is part of the protein resistant family to multi drugs MDR [57]. So far, these mechanisms would be exclusively of *C. albicans*; therefore, it cannot be extrapolated to other organisms. Nevertheless, it is necessary to carry out more studies on other yeasts, with other peptides and other media to determine the existence of similar mechanisms and/or new resistant mechanisms to AMPs.

## 8. Conclusions

*B. bruxellensis* is the most important spoilage yeast in wine at a world level, due to the negative sensorial effect when it is present. It has been studied using different methodologies to eradicate its presence in wine, but the cost and efficiencies of these has meant that, actually, in the wine

industry, $SO_2$ is still used as a microbial controller. However, the $SO_2$ presence in wine can bring health problems; so, it is necessary to search for natural products that allow for the control of microbial growth, especially spoilage microorganisms. The AMPs are peptides of low molecular mass that are secreted by microorganisms ensuring survival. Our work group identified that *C. intermedia* LAMAP1790 secretes peptides with antimicrobial activity, which have a molecular mass lower than 10 kDa. These peptides control the *B. bruxellensis* growth without affecting *S. cerevisiae*. However, it is necessary to determine their action at pH variation, residual sugar concentration, the increase of the ethanol concentration, and proliferation of other yeasts and/or related bacteria in winemaking to define the industrial potential of these AMPs. Likewise, the low production of these peptides is a problem, it being necessary to develop biotechnological tools that allow larger production of these peptides and thus enable the carrying out of assays at an industrial scale.

**Author Contributions:** Conceptualization, R.P. and M.A.G.; methodology, R.P, R.C., and M.A.G.; writing—review and editing, R.C, A.R, R.P., and M.A.G.

**Funding:** This research was funded by grants Dicyt 081871GM_DAS, Proyecto Fortalecimiento Usach USA1398_GM181622 and the Comisión Nacional de Investigación Científica y Tecnológica (CONICYT) PCHA/DoctoradoNacional/2013-21130439 doctoral fellowship.

**Conflicts of Interest:** The authors declare no conflicts of interest.

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
