# Peer review of "A Control Alternative for the Hidden Enemy in the Wine Cellar"

_fermentation, doi:10.3390/fermentation5010025_

Round 1
Reviewer 1 Report
This is a very interesting topic and a state-of-the-art review is appropriate at this point of knowledge. However the title seems to be a bit misleading. Most of the review is about antimicrobial peptides and their mode of action with relatively little work done on Brettanomyces. In particular, the authors do not present evidence that peptides are an efficient tool against this yeast under realistic wine conditions. Otherwise it seems that AMP are already well established as an alternative to sulfite or other antimicrobials.
Specific corrections
Line 11 – replace must by wine because musts are seldom highly contaminated by Brettanomyces and do not require its inactivation during fermentation, only later after alcoholic fermentation.
Figure 1 – remove this figure because it is already disseminated in many other references.
Section 3 – there is little revision of the control measures, like DMDC, chitosan, or thermal treatments. Authors may refer to several recent reviews on Brettanomyces control (e.g. wrtitten by the research groups of C. Edwards or A. Morata), where one last has been published in one MDPI jornal (Beverages, by Malfeito-Ferreira, 2018).
Line 98 and 178 – correct the font of the enzyme words and of the website.
Line 175 – correct (me parece).
Line 180 – spell AFP.
Line 196 – spell PAF.
Figure 3A – the c columns do not have te top line.
Tables 1 – too little information, remove and replace by text only.
Conclusions: - correct to Conclusions
Conclusions – authors are invited to increase the text mainly concerning by one hand the limitations found so far and by the other hand the future prospects to exploit the potential of AMP against Brettanomyces in wine. In fact, AMP’s seem to be far from an industrial application mainly because it is not easy to overcome the difficulties of peptide purification and mode of use.
Author Response
Dear Reviewer
We are grateful to the comments done about our manuscrip. Accordong your correction, we made a major revision.
Thanks for your careful review.
Question | Response |
Line 11 – replace must by wine because musts are seldom highly contaminated by Brettanomyces and do not require its inactivation during fermentation, only later after alcoholic fermentation | It was done
|
Figure 1 – remove this figure because it is already disseminated in many other references. | In the new version it was removed. |
Section 3 – there is little revision of the control measures, like DMDC, chitosan, or thermal treatments. Authors may refer to several recent reviews on Brettanomyces control (e.g. written by the research groups of C. Edwards or A. Morata), where one last has been published in one MDPI jornal (Beverages, by Malfeito-Ferreira, 2018). | In the new version was done: line 117-127 and 134-136. It was added new references 12 and 21. |
Line 98 and 178 – correct the font of the enzyme words and of the website | It was done. |
Line 175 correct (me parece). | It was done |
Line 180 – spell AFP | It was done, line 197 |
Line 196 – spell PAF | It was done, line 209 |
Figure 3A – the c columns do not have the top line. | In the new version the Figure 3 was removed (according the Reviewer 2) |
Tables 1 – too little information, remove and replace by text only | In the new version this table was removed (according the Reviewer 2) |
Conclusions: - correct to Conclusions
| The conclusion was re-written |
Reviewer 2 Report
The manuscript is submitted as a review paper. Most part of the manuscript is reviewing the literature (lines 27-276) and then it alters to a research paper format reporting new data.
In brief, the manuscript is a review or a research one? Three quarters of it are reviewing the literature and suddenly in page 6, line 258 it starts reporting new data. This is an unacceptable format.
In addition the manuscript requires very good english editing in order to become understandable to the reader.
The manuscript should be rejected and re-wrtitten either as review or with sustantianl amount of data as research paper
Author Response
Dear Reviewer:
We are grateful to the comments done about our manuscript. According your correction, we made a major revision.
Question | Response |
The manuscript is submitted as a review paper. Most part of the manuscript is reviewing the literature (lines 27-276) and then it alters to a research paper format reporting new data. In brief, the manuscript is a review or a research one? Three quarters of it are reviewing the literature and suddenly in page 6, line 258 it starts reporting new data. This is an unacceptable format. | In the new version it was corrected |
In addition the manuscript requires very good english editing in order to become understandable to the reader. | The manuscript was revised by a native speaker English |
The manuscript should be rejected and re-wrtitten either as review or with substantial amount of data as research paper
| In the new revision it was done |
Thanks for your careful review.

Reviewer 3 Report
The review "A Control Alternative for the Hidden Enemy in the Wine Cellars" by Pena et al. is timely and should be well received by the technical fermentation community. The text is written with wine as the focus, but indeed, the information is applicable to other fermented beverages as well. The authors also supply some new data (Fig.3 and Table 1) to bolster their recently published findings (citation 51). It is unclear if Fermentation allows such non-peer reviewed data - I cannot perform an adequate review here due to the lack of a methods section - but it only accounts for a small percentage of the overall review regardless.
My main issues were: 1) it needs quite a bit of English language editing, and 2) some (ostensibly) new data are presented. It's unclear to me if a review can include unpublished data. I believe that this manuscript is suitable for publication.
Author Response
Dear Reviewer:
We are grateful to the comments done about our manuscript. According your correction, we made a major revision.
1. Question: The review "A Control Alternative for the Hidden Enemy in the Wine Cellars" by Pena et al. is timely and should be well received by the technical fermentation community. The text is written with wine as the focus, but indeed, the information is applicable to other fermented beverages as well. The authors also supply some new data (Fig.3 and Table 1) to bolster their recently published findings (citation 51). It is unclear if Fermentation allows such non-peer reviewed data - I cannot perform an adequate review here due to the lack of a methods section - but it only accounts for a small percentage of the overall review regardless.
Response: In the new version it was corrected.
2. Question: My main issues were: 1) it needs quite a bit of English language editing, and 2) some (ostensibly) new data are presented. It's unclear to me if a review can include unpublished data. I believe that this manuscript is suitable for publication.
Response: In the new version it was corrected, and it was revised by a native speaker English
Thanks for your careful review.
Round 2
Reviewer 1 Report
Authors made the required ammendments
Reviewer 2 Report
I am puzzled because it is the first time in my scientific career I am facing with this type of dilemmas. I think the editor should send the revised version to the acaedmic editor. As long as the Academic Editor is ok by the manuscript I am ok as well.